# Demographic and socioeconomic patterns in healthcare-seeking behaviour for respiratory symptoms in England: a comparison with non-respiratory symptoms and between three healthcare services

Kirsty E Morrison [1], Felipe J Colón-González [1,2], Roger A Morbey [3], Paul R Hunter [4], Judith Rutter,[5] Gareth Stuttard,[6] Simon de Lusignan [7,8], Alex Yeates,[9] Richard Pebody [10], Gillian Smith [1,3], Alex J Elliot [3], Iain R Lake [1]

For numbered affiliations see end of article.

**Correspondence to**
Kirsty E Morrison;
k.morrison@uea.ac.uk

## ABSTRACT

**Objective** This study will analyse respiratory contacts to three healthcare services that capture more of the community disease burden than acute data sources, such as hospitalisations. The objective is to explore associations between contacts to these services and the patient's age, gender and deprivation. Results will be compared between healthcare services, and with non-respiratory contacts to explore how contacts differ by service and illness. It is crucial to investigate the sociodemographic patterns in healthcare-seeking behaviour to enable targeted public health interventions.

**Design** Ecological study.

**Setting** Surveillance of respiratory contacts to three healthcare services in England: telehealth helpline (NHS111); general practitioner in-hours (GPIH); and general practitioner out of hours unscheduled care (GPOOH).

**Participants** 13 million respiratory contacts to NHS111, GPIH and GPOOH.

**Outcome measures** Respiratory contacts to NHS111, GPIH and GPOOH, and non-respiratory contacts to NHS111 and GPOOH.

**Results** More respiratory contacts were observed for females, with 1.59, 1.73, and 1.95 times the rate of contacts to NHS111, GPOOH and GPIH, respectively. When compared with 15–44 year olds, there were 37.32, 18.66 and 6.21 times the rate of respiratory contacts to NHS111, GPOOH and GPIH in children <1 year. There were 1.75 and 2.70 times the rate of respiratory contacts in the most deprived areas compared with the least deprived to NHS111 and GPOOH. Elevated respiratory contacts were observed for males <5 years compared with females <5 years. Healthcare-seeking behaviours between respiratory and non-respiratory contacts were similar.

**Conclusion** When contacts to services that capture more of the disease burden are explored, the demographic patterns are similar to those described in the literature for acute systems. Comparable results were observed between respiratory and non-respiratory contacts

## Strengths and limitations of this study

► One of the largest multiservice, observational studies ever undertaken of over 13 million respiratory-related contacts to explore the demographic and socioeconomic patterns of healthcare-seeking behaviour for respiratory symptoms in England.

► Data from three commonly used community healthcare services: telehealth helpline (NHS111); general practitioner in-hours (GPIH); and general practitioner out of hours unscheduled care (GPOOH), were included in the study, allowing for a comprehensive analysis of respiratory healthcare-seeking behaviour in the community.

► For NHS111 and GPOOH services, non-respiratory contacts were explored to allow us to identify similarities and differences between demographic and socioeconomic patterns in healthcare-seeking behaviours for respiratory and non-respiratory illnesses.

► Data were obtained from the real-time surveillance of three community healthcare services and due to this did not contain information of comorbidities. Therefore, this could not be accounted for in the analysis.

suggesting that when a wider spectrum of disease is explored, sociodemographic factors may be the strongest influencers of healthcare-seeking behaviour.

## INTRODUCTION

Acutely presenting respiratory diseases, henceforth referred to as respiratory disease, including upper and lower respiratory tract infections (URTI and LRTI) and asthma, have a substantial impact on individual health

and healthcare systems. Globally, LRTI were the sixth leading cause of death, causing over 2.3 million deaths in 2016.[1] URTI have a substantial impact on health burden with over 4.7 million disability-adjusted-life years (DALYs) globally in 2016.[2] Although asthma is a chronic condition, it often presents with acute exacerbations. Worldwide, as of 2015, around 358 million people were living with asthma.[3]

In the Global Burden of Disease Study 2010,[4] the UK had the second highest number of age-standardised DALYs due to LRTI and asthma out of 19 other high-income countries.[5] In comparison to 15 other European countries, the UK had one of the highest mortality rates due to respiratory infections.[6] In the UK, 15.8% of the population are predicted to develop asthma in their lifetime.[6] LRTI and URTI are estimated to cost the UK over £1.7 billion, and asthma over £3 billion annually.[7]

To help develop effective interventions, it is necessary to understand who is most at risk. Deprivation has been linked to morbidity and mortality of asthma and respiratory infections.[8–11] Factors attributed to higher rates of respiratory diseases in more deprived areas include higher smoking rates,[12] higher levels of pollution[13] and poor-quality housing.[14] Females have higher rates of presentation with URTI compared with males.[15] However, males succumb more to LRTI, which are more severe than URTI and lead to higher mortality rates.[15] Gender differences in respiratory diseases are affected by age, with male children more prone to illness, hospitalisation, and death due to respiratory diseases.[16] This age-gender interaction has been observed in asthma, with prepubescent boys more likely to develop asthma; by early puberty the prevalence equalises.[17–19]

Here, we focus on the sociodemographic patterning of respiratory disease in England. Respiratory diseases can often be self-limiting, and therefore may be under-reported in national surveillance from traditional data sources such as laboratory reports and hospitalisations. This can lead to bias in the reported relationship between healthcare-seeking behaviour for respiratory diseases, and sociodemographic factors. Unlike previous studies, which often focus on traditional data sources (eg, laboratory reports, hospitalisations), we use data sources from telehealth, family doctors and unscheduled care that may provide a more complete reflection of community burden. The data used in this study are defined as non-specific, prediagnostic-syndromic data which we use as a proxy for disease.[20]

Syndromic data have previously been used to investigate associations between demographics, deprivation and disease. Todkill et al,[21] used syndromic data from family doctors' presentations for allergic rhinitis to investigate sociodemographic associations. Higher rates of allergic rhinitis were observed in females, children and those from more deprived areas. Lusignan et al,[22] used family-doctor coded diagnoses to conduct a large-scale investigation on the impact of age, gender and deprivation on respiratory illnesses in England. Those in the most deprived quintile had a higher probability of presenting with common cold. Nilsson et al,[23] investigated associations between antibiotic prescriptions for penicillin-non-susceptible Streptococcus pneumonia (PNSP) and deprivation in Sweden. Although deprivation was not linked with higher rates of PNSP, higher deprivation was associated with increased rates of antibiotic prescribing.

One issue with the use of syndromic data in such studies is the difficulty in ascertaining whether associations arise from healthcare-seeking behaviours or disease incidence. This situation could be explored by comparison of results to all healthcare-seeking behaviours, but this appears absent in the literature. Furthermore, most studies use a single source of syndromic data making it difficult to know the generability of results.

### Aims
This study aims to:
1. Explore associations between respiratory-related contacts to three community healthcare services; telehealth, family doctors and unscheduled care, and age, gender and ecological measures of deprivation in England.
2. Compare these results to all non-respiratory contacts to identify whether associations are specific to respiratory disease.

## METHODS
### Data collection
#### Syndromic data
Public Health England (PHE) coordinates a national programme of syndromic surveillance of multiple healthcare services.[24] Our study uses routinely available syndromic data from a telehealth service (NHS111) which operates continuously all year; a family doctor service (general practitioner in-hours; GPIH) that operates during weekday working hours, and an out-of-hours family doctor service (general practitioner out of hours; GPOOH) (table 1). These services run as part of the National Health Service (NHS) which is universal and free at the point of delivery.

Syndromic data, obtained from the three surveillance systems coordinated by PHE, comprised annual counts, between 1 January 2015 and 31 December 2016, of respiratory and non-respiratory contacts. Syndromic indicators classified as respiratory for this study are presented in table 1 (and described in online supplemental appendix 1). Previous studies have demonstrated the association between acute respiratory diseases and these syndromic indicators.[25 26] Based on expert knowledge these indicators were chosen to characterise acutely presenting respiratory illnesses such as asthma, and respiratory infections, and to be as comparable as possible between the three healthcare services. Respiratory contacts for each service comprised the sum of the respiratory indicators listed in table 1. Non-respiratory counts comprised the difference between the total number of contacts and the number

**Table 1** Indicators for respiratory diseases for each syndromic surveillance system, 1 January 2015 to 31 December 2016

| Healthcare service | Healthcare service provided | Coding system for healthcare service | Contact type | Routine syndromic indicator included for this study | Number of contacts used in study |
|---|---|---|---|---|---|
| NHS111 | NHS111 is a free non-emergency medical helpline. It operates 24/7 and is staffed by trained call handlers. A clinical decision support system is used to structure the response to the call, with the call disposal ranging from advice about self-care to dispatch of an emergency ambulance. | NHS Pathways[43] | Acute respiratory calls | 'Cold/influenza', 'cough' and 'difficulty breathing' | 1 721 034 |
| | | | Total number of calls | All contacts | 21 242 154 |
| | | | Non-acute respiratory calls | All non-acute respiratory contacts (All contacts − acute respiratory contacts) | 19 521 120 |
| GPIH | GPIH are primary care services that provide free scheduled day-to-day healthcare in England. General practitioners treat all common medical conditions and depending on the condition will refer patients to hospitals and other medical services for urgent and specialist treatment. | Read codes v2 (hierarchical) and v3 (non-hierarchical). Full description in Robinson et al[44] | Acute respiratory consultations | 'Upper and lower respiratory tract infections' and 'asthma' | 10 310 626 |
| | | | Total number of consultations | Not available | Not available |
| | | | Non-respiratory consultations | Not available | Not available |
| GPOOH | GPOOH services provide free access to primary healthcare when GPIH services are closed, which is typically weekdays 18:30 to 08:00, weekends and bank holidays. | Read codes[44] | Acute respiratory consultations | 'Acute respiratory infection' 'difficulty breathing/wheeze/ asthma' | 1 562 883 |
| | | | Total number of consultations | All contacts | 8 500 540 |
| | | | Non-acute respiratory consultations | All non-acute respiratory contacts (All contacts − acute respiratory contacts) | 6 937 657 |

GPIH, general practitioner in hours; GPOOH, general practitioner out of hours; NHS, National Health Service.

of respiratory contacts. The total number of contacts was not available through the surveillance system of GPIH, and therefore non-respiratory counts were not available.

Data were obtained at the finest geographical level available; postcode district (PD) (eg, SW1) for NHS111 and GPOOH, and PHE Centre (eg, London) for GPIH. In England, there are 2 234 PD with, on average, 25 660 (range: 142–162 266) residents. There are nine PHE Centres with an average of 6 181 375 (range: 2 644 727–9 080 825) residents in England. Count data were subdivided by age group: <1, 1–4, 5–14, 15–44, 45–64, 65–74 and >75 years; by gender (male/female), year (2015/2016); and by geographical location.

### Demographic data

PD population, obtained from the 2011 census,[27] grouped by age and gender, was used as a denominator in the analysis of NHS111 and GPOOH contacts. GPIH populations were derived from the sum of registered populations at each participating GPIH practice.

### Independent variables

The Index of Multiple Deprivation (IMD)[28] was used as an area-level measure of deprivation. This index in calculated from seven domains; income, employment, education, health (premature death and poor physical/mental health), crime, barriers to housing and services, and living environment. This index was used, rather than more specific variables (eg, smoking behaviour) to avoid issues with collinearity. IMD scores were obtained at lower layer super output area (LSOA); a weighted mean for each PD was calculated using the proportion of LSOA population in each PD. Weighted means were divided into quintiles from most (1) to least (5) deprived.

### Data cleaning and exploration

NHS111 respiratory contacts from Essex in 2015 and 2016 and Norfolk in 2016 were excluded because syndromic data were unavailable. Syndromic surveillance coverage maps of GPOOH were obtained at the upper tier local authority (UTLA) geographical level, and data were excluded from any UTLA where the PHE surveillance programme received little or no syndromic data.

For all systems, data were excluded if location, age or gender of healthcare seeker was unknown (table 2). PD that were demarcated for large organisations (eg, Heathrow Airport), or had less than 200 residents were excluded from analysis as their small populations or unique nature are unlikely to produce reliable estimates. PD that overlap borders with Scotland and Wales were excluded.

For each system, contact rates were mapped for both study years to visualise spatial variation of the data (online supplemental appendix 2).

### Statistical analysis

To measure the relationship between the dependent and independent variables generalised linear mixed models (GLMMs) were used for NHS111 and GPOOH, while a generalised linear model (GLM) was used for GPIH. GPIH data were modelled using both GLM and GLMM methods; however, the GLM provided a better model fit (online supplemental appendix 3).

The variables of interest for analysis of NHS111 and GPOOH data were age, gender and deprivation. Two-way interactions between age and gender, and age and deprivation were also investigated. A categorical variable for year was included in the model to account for interannual variation. Percentage urban area was included to account for differences in healthcare-seeking behaviour or disease risk related to urbanicity. Due to the highly aggregated deprivation and percentage urban data at PHE centre level, these variables were not included in the analysis of GPIH data, as non-significant estimates would likely be due to the lack of variation within this aggregated data. Therefore, the variables of interest, age and gender and their interactions were investigated in the analysis of GPIH data. To account for population differences at the geographical level, the logarithm of the population plus one was included as a model offset.

| Table 2 | Number of contacts excluded from study by reason for exclusion | |
| --- | --- | --- |
| | **NHS111 (% of total)** | **GPOOH (% of total)** |
| Total number provided | 21 905 099 | 9 623 939 |
| Reason for exclusion | | |
| No valid postcode provided/not in England | 613 495 (2.9) | 92 815 (9.6) |
| No gender given | 9 536 (0.04) | 12 312 (0.13) |
| No age given | – | 1 601 (0.017) |
| Postcode district with <200 population | 44 (0.0002) | 12 (0.0001) |
| Overlapping borders with Scotland or Wales | 17 072 (0.08) | 3 795 (0.04) |
| Large area users/city centres | 1 517 (0.007) | 2 379 (0.03) |
| Poor coverage/data issues | 21 281 (0.1) | 1 010 485 (10.50) |
| Total excluded | 662 945 (3.02) | 1 123 399 (11.67) |

*General practitioner in hours had no exclusions.
GPOOH, general practitioner out of hours.

The study design accounted for the hierarchical structure of the data for each system by including PD and UTLA as random effects in the NHS111 and GPOOH models. UTLA was included as a random effect to account for similar characteristics of neighbouring PD to reduce the effect of spatial autocorrelation, and because UTLA are responsible for services that might influence contact rates (map of UTLA overlaid PD in online supplemental appendix 4). Where a PD was located in more than one UTLA, the largest PD area was allocated. PHE Centre was included as a fixed effect in the GPIH model.

We explored Poisson and negative binomial model specifications to account for potential overdispersion in the data. Overdispersion was tested by comparing the sum of squared Pearson residuals to the residual degrees of freedom. Models with overdispersion statistics <1.5 were deemed acceptable.[29] Wald tests were used to determine the overall significance of variables. The algebraic definition of the models is described in online supplemental appendix 5.

Model overfitting and the predictive ability of the model was assessed using k-fold cross validation where the data were split into 10 equal groups (k). Each group was used to train the model k−1 times and test the model once. To assess the model, mean absolute error was used and the results are presented in online supplemental appendix 6. Rate ratios (RR) with 95% CIs were estimated for the main effects: age, gender and deprivation. To allow the visualisation of the main and interaction effects, and comparisons of trends between the contact types and services, the number of contacts to each service were predicted using the models, then standardised and plotted. The predictions were standardised to a zero mean and unit variance by subtracting the mean of the predictions from each predicted value and then dividing by the standard deviation (SD).

All analyses were conducted in R V.3.5.2[30] and models were specified using the glmmTMB[31] and MASS[32] packages.

## RESULTS
### Data and model selection
Table 2 indicates that relatively few contacts were excluded from analysis due to unknown location, age or gender, with only 3.02% of NHS111 and 11.67% of GPOOH data excluded. GPIH data had no exclusions because patients that used this service have to preregister and therefore location, age and gender are known. Any data issues observed in the datasets were because of the passive reporting to the surveillance systems and not due to disruption of healthcare services. Table 2 demonstrates that in total 21 242 154 contacts to NHS111 were included in the analysis, of which 8.10% (n=1 721 034) were respiratory contacts; 6 937 657 GPOOH contacts were included in the analysis of which 22.53% (n=1 562 883) were respiratory contacts. The different proportions of respiratory contacts between the NHS111 and GPOOH

services likely reflect to the different functions of the services, and the severity of illness for which each service would be contacted by patients. Total number of contacts were not available for GPIH, but 10 310 626 respiratory contacts were included in the analysis of GPIH data.

Two model distributions were considered for analysis: Poisson, and negative binomial, with models selected by considering overdispersion of the data. Negative binomial models handled the overdispersed data best in all five models (descriptions of model fit in online supplemental appendix 7).

Overall, the models performed well with low mean of mean absolute error values (online supplemental appendix 6). The fixed effects explained a high amount of variation in both NHS111 and GPIH respiratory models, with a marginal $R^2$ of 0.86 and an $R^2$ of 0.99, respectively (online supplemental appendix 7). The fixed effects explained less of the variation in the GPOOH respiratory model, with a marginal $R^2$ of 0.30. When the spatial random effects were considered in the NHS111 and GPOOH respiratory models, both models had a conditional $R^2$ of over 0.94. In the GPOOH models, the difference observed between marginal and conditional $R^2$ values compared with the NHS1111 models is because less of the variation in the data can be explained by the fixed effects. The difference between the marginal and conditional $R^2$ in the GPOOH models is likely due to the uncertainty of the underlying study population, with this uncertainty explained by the geographical levels included as random effects (random effects of PD and UTLA).

### Multivariable analysis
Standardised predictions, the number of SD from the mean, of each model are visualised in figures 1–3 alongside overall significance of each model. The table of main effect RR and 95% CI is presented in online supplemental appendix 7.

Respiratory contacts in the <1 year age group are significantly higher compared with the reference group (15–44 years) in all three services (online supplemental appendix 7), with 37.32 (95% CI 36.10 to 38.85), 18.66 (95% CI 17.78 to 19.58) and 6.21 (95% CI 5.96 to 6.48) times the rate of contacts to NHS111, GPOOH and GPIH, respectively. The comparative differences between respiratory contacts in the <1 year age group and the reference group was highest for NHS111 compared with GPOOH and GPIH. Although contacts are highest in the <1 year age group compared with the reference group in the non-respiratory models, similar RR were observed between the services; with 7.03 (95% CI 6.87 to 7.20), and 7.64 (95% CI 7.28 to 8.02) times the rate of non-respiratory contacts to NHS111, and GPOOH. This relationship is visualised in figure 1A

Gender has a significant influence on contacts to each service (online supplemental appendix 7); visually, the trend appears similar between respiratory and non-respiratory contacts and across all services (figure 1B). There are 1.59 (95% CI 1.56 to 1.62), 1.73 (95% CI 1.70 to 1.77) and 1.95 (95% CI 1.87 to 2.03) times the rate

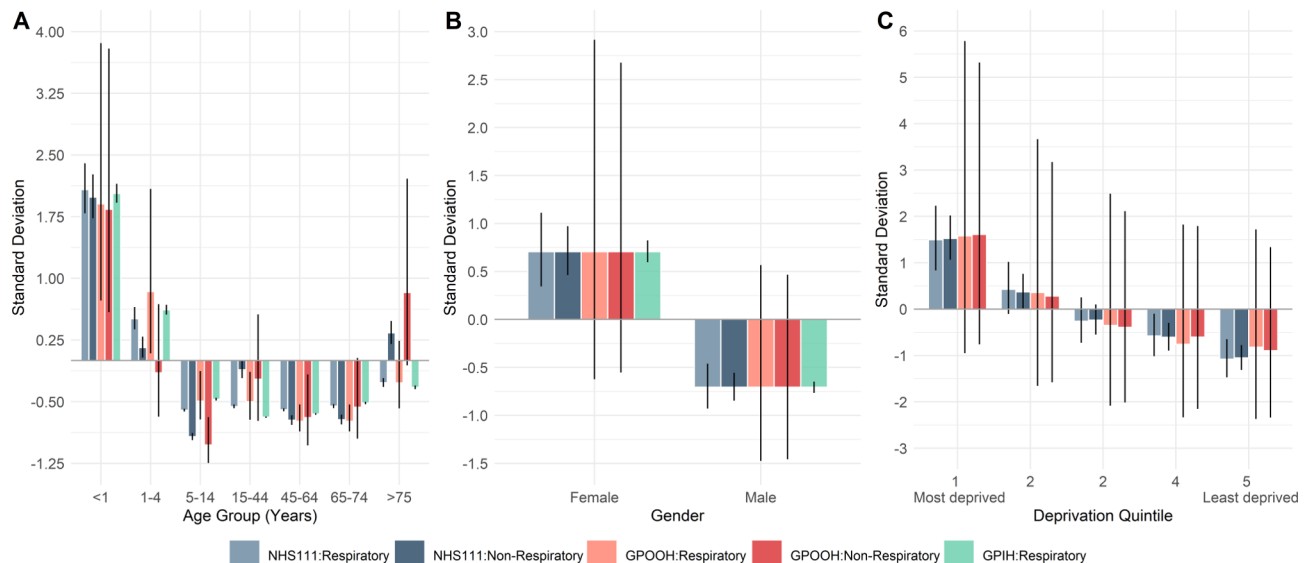

**Figure 1** Standard deviation from the mean of main effects predictions from the multivariable analysis of respiratory contacts to NHS111, general practitioner out of hours (GPOOH) and general practitioner in hours (GPIH): each plot describes the SD of (A) age group, (B) gender and C) deprivation. All models had an overall significance ≤0.001.

of respiratory contacts regarding females to NHS11, GPOOH and GPIH, respectively.

Deprivation is significant in both NHS111 models (online supplemental appendix 7), which found there are 1.75 (95% CI 1.56 to 1.95) times the rate of respiratory contacts in the most-deprived areas compared with the least-deprived (IMD quintile 1 vs 5), and 1.81 (95% CI 1.66 to 1.99) times the rate of non-respiratory contacts. Deprivation was significant in both the respiratory and non-respiratory GPOOH models, with 2.70 (95% CI 1.79 to 4.08) times the rate of respiratory contacts and 2.70 (95% CI 1.89 to 3.85) times the rate of non-respiratory contacts in most-deprived areas compared with least-deprived. Similar RR estimates for deprivation were

observed in respiratory and non-respiratory NHS111 and GPOOH contacts. This relationship is visualised in figure 1C, where the similarities between NHS111 and GPOOH, and respiratory and non-respiratory contacts can be observed.

Age–gender interactions show similar trends across system types and when compared with non-respiratory contacts. Overall, there are more female contacts; but in all models (figure 2), in the <1 and 1–4 age groups there were more contacts regarding males.

Age–deprivation interactions were investigated for NHS111 and GPOOH. In both respiratory and non-respiratory contacts to NHS111 (figure 3A), the trends suggest there are more contacts in the most deprived

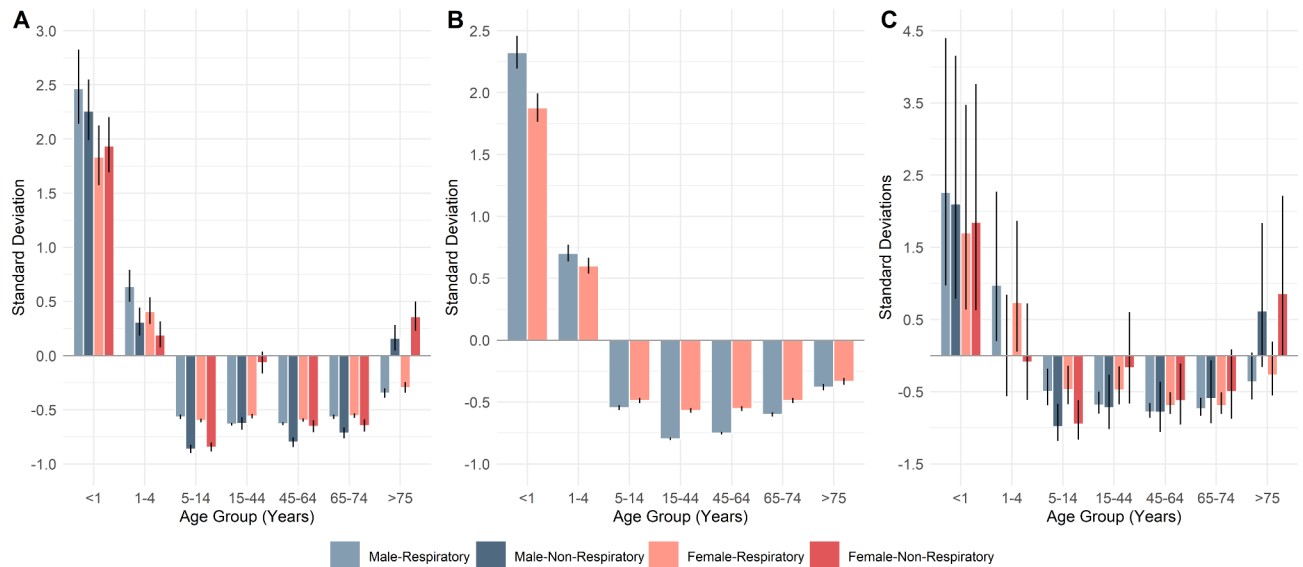

**Figure 2** Standard deviation from the mean of age gender interactions for respiratory and non-respiratory contacts by each system: (A) NHS111, (B) general practitioner in hours (GPIH) and (C) general practitioner out of hours (GPOOH). All models had an overall significance ≤0.001.

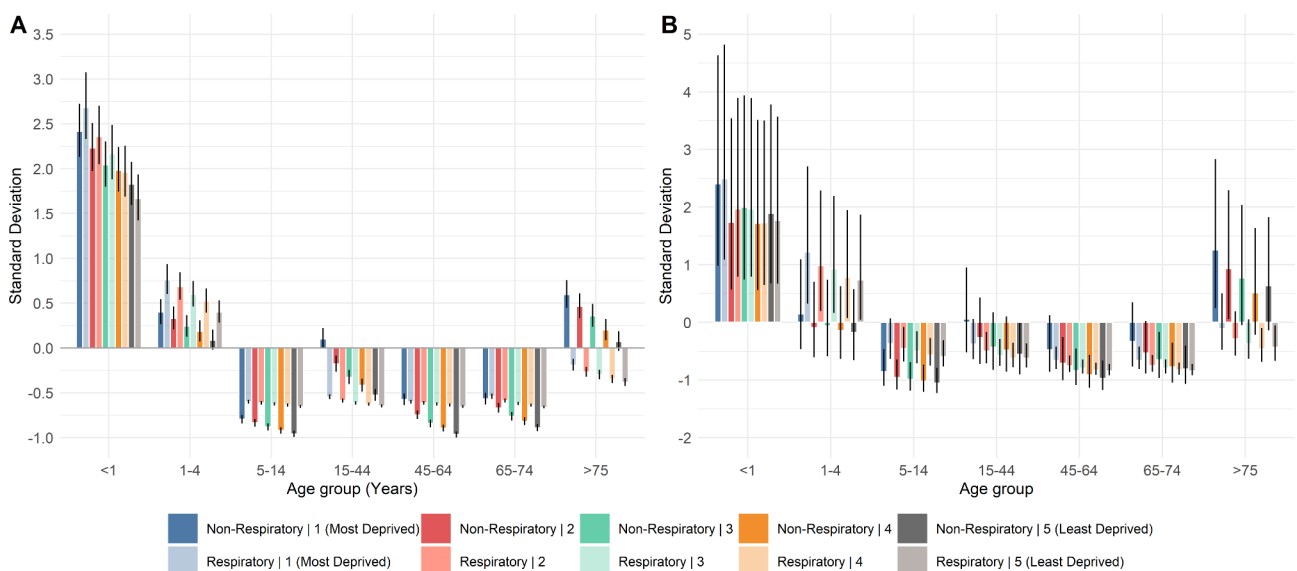

**Figure 3** Standard deviation from the mean of age deprivation interactions for respiratory and non-respiratory contacts by each system: (A) NHS111 and (B) general practitioner out of hours (GPOOH). NHS111 respiratory, GPOOH respiratory and GPOOH non-respiratory models had an overall significance ≤0.001. NHS111 non-respiratory model had an overall significance of ≤ 0.01.

quintiles across all ages. For respiratory contacts, there is a stronger trend with deprivation in the <1, 1–4 and >75 year age groups, and a weaker trend in the remaining age groups. This NHS111 trend is strong across all age groups for non-respiratory contacts, particularly in the age group >75 years. There was a similar linear trend with deprivation and age for respiratory and non-respiratory contacts to GPOOH (figure 3B), with more contacts in the most-deprived areas in all age groups.

## DISCUSSION

### Impact

We present a large-scale analysis of over 13 million respiratory-related contacts to three healthcare services in England. These services deliver healthcare to patients at the community level and may be better indicators of overall disease patterns as opposed to the relatively small proportion that appear in hospitalisation or laboratory-based surveillance. Access to these large syndromic datasets has allowed a comprehensive analysis of the demographics using of each service and the impact deprivation has on healthcare contacts. It has also allowed the interactions between these factors to be explored. By analysing respiratory and non-respiratory contacts, we were able to identify differences and similarities in usage patterns. Analysing NHS111 and GPOOH contacts at PD, a small level of geography, we explored deprivation patterns to give a thorough analysis of social patterning of factors associated with respiratory contacts.

### Main findings and comparison to literature

In all three services (NHS111, GPOOH and GPIH), there were more respiratory and non-respiratory contacts in females. Previous research indicates that women are

more likely to seek healthcare than men, even when female-specific concerns are accounted for.[33 34] Although we observed higher respiratory contact rates for in females, males have been observed to have higher death rates due to respiratory disease,[9] highlighting possible gender differences in healthcare-seeking behaviour or sex differences in severity of respiratory disease. In terms of age, contacts were highest in <1, followed by the 1–4 and the >75 age groups. These trends were observed across all three services, and fit with previous research.[16 35] Although the three services displayed similarities in terms of the age groups more likely to use the service, there were differences in the magnitude of the RRs. In comparison to the reference group (15–44 years) children <1 year were 37.3, 18.7 and 6.2 times more likely to contact NHS111, GPOOH and GPIH, respectively, due to respiratory disease. These differences could reflect the routes to access to the different services (table 1), and the severe and sudden nature of the respiratory illness in very young children requiring urgent health advice.

When interactions between age and gender were explored, we found that more contacts were regarding males in the two youngest age groups for both respiratory diseases; this effect was observed in all services. Previous studies have reported that male children have higher rates of respiratory illnesses.[16 17 36] Lusignan et al,[36] observed a higher incidence of family doctors contacts in males <15 years due to LRTI compared with females of the same age, with the largest gender difference in the youngest age group (<1 year). This study also noted a higher contact rate in males <15 years due to asthma. The reasons for which are unclear, but immunological, genetic and biological differences are thought to increase risk, suggesting that the excess in male children is due to

genuine predisposition to respiratory illness rather than sociological factors.[16] This excess in male children was also observed in our data for non-respiratory contacts. The reasons for this excess of non-respiratory contacts in male children is unclear. These findings suggest that there are similar drivers to contacts in male children presenting for all illness types rather than with just respiratory diseases, whether this be due to differences in healthcare-seeking behaviours or disease aetiology. Earp *et al* observed that adults perceived that male children (aged 5 years) experienced more pain compared with females of the same age, despite the same clinical circumstances and identical pain behaviour.[37] The Infectious Intestinal Disease (IID) Study observed a higher rate of IID in the community in females <1 year compared with males, but a higher rate of contacts to family doctors of males compared with females of the same age.[38] This observation suggests that either males were more likely to have cases of IID, which required medical intervention, or healthcare was more likely to be sought for males compared with females of the same age. There could be gender differences in the perception of ill health in young children from parental caregivers, which could influence healthcare-seeking behaviour, although this requires more research.

Deprivation was significant in both the NHS111 and GPOOH models, with a greater risk of respiratory and non-respiratory contacts in those from more deprived areas. Similar results were found when looking at gastro-intestinal illness contacts to NHS111 and deprivation, with more contacts in the most deprived areas.[39] Kelly *et al*,[40] observed higher attendances from those living in the most deprived areas to GPOOH services. The findings observed in this study are similar to those described in the literature from more acute healthcare settings, with previous studies linking increased deprivation with increased rates of hospitalisations and death due to respiratory infection.[9 41 42]

One of the relatively unique elements of our work was the exploration of the interactions between deprivation and age. For NHS111, when stratified by age, deprivation appears to have a greater impact on non-respiratory contacts than on respiratory contacts. The exception is in the youngest age bracket, where deprivation has a similarly high impact on both respiratory and non-respiratory contacts, with more contacts in most-deprived areas. A similar pattern emerged for GPOOH where there was more contact in the most derived areas; however, this trend was similar across age groups and in respiratory and non-respiratory contacts.

It is important to consider how these results can be generalised outside of England. It is difficult to make direct comparisons to other countries due to differences in healthcare systems, public health surveillance infrastructure, population demographics and deprivation levels. With the rise internationally in use of syndromic and non-traditional forms of surveillance (such as those used in this study), we feel it is important for other counties who use these forms of surveillance to undertake similar research.

## Limitations

Although the surveillance of these three services allows us to observe a large number of healthcare contacts, these are working surveillance datasets from real-time surveillance system, with periods where data are not received. The large numbers included in the study meant that socioeconomic and demographic patterns could still be observed, and omissions are only likely to bias estimates if correlated with independent variables. Coverage issues will increase the uncertainty of our estimates, which is particularly evident in the wide CIs observed in the GPOOH results.

The analysis presented here is exploring indicators of disease as opposed to actual disease. Therefore, the analyses are reliant on accurate classifications of disease indicators. NHS111, GPIH and GPOOH services have different symptom coding systems, and although we tried to choose indicators which would allow comparisons, the definition of respiratory and non-respiratory may differ between services. The deprivation measures used were composites of multiple factors that contribute to deprivation and were ecological in nature, are subject to the ecological fallacy whereby the associations found at the area level may not hold at the individual level. Data obtained from the surveillance systems did not contain information on comorbidities or ethnicity; therefore, these factors could not be accounted for in the analysis. Data from each surveillance system were available in an aggregated format, and hence it was not possible to identify multiple contacts by the same patient to the services during the study period.

## CONCLUSION

This large-scale study highlights the effects of age, gender and deprivation on healthcare-seeking behaviours for respiratory diseases in the community, with more healthcare contacts from females, the young and old and those from more deprived areas. Similar patterns were observed across the three services, which were in agreement with the literature. Comparable results were also observed between respiratory and non-respiratory contacts suggesting that even when a wider spectrum of disease is explored, demographic and socioeconomic factors may be the strongest influencers of healthcare-seeking behaviours. When broken down by age, there were more contacts regarding male children <5 years compared with females. This trend was observed in both respiratory and non-respiratory contacts indicating there are more healthcare-seeking behaviours in male children across a range of disease types. These findings could be influenced by sociological factors as well as disease aetiology and requires further research. Further research is required to understand the role that age, gender and deprivation has on healthcare-seeking behaviours.

**Author affiliations**
[1]School of Environmental Sciences, University of East Anglia, Norwich, UK

[2]Centre for Mathematical Modelling of Infectious Diseases, London School of Hygiene and Tropical Medicine, London, UK
[3]Real-time Syndromic Surveillance Team, Field Service, National Infection Service, Public Health England, Birmingham, UK
[4]Norwich Medical School, University of East Anglia, Norwich, UK
[5]NHS Digital, Leeds, UK
[6]NHS England and NHS Improvement, London, UK
[7]Nuffield Department of Primary Care Health Sciences, University of Oxford, Oxford, UK
[8]Royal College of General Practitioners Research and Surveillance Centre, London, UK
[9]Advanced, Ashford, UK
[10]Influenza and Other Respiratory Virus Section, Immunisation and Countermeasures Division, National Infection Service, Public Health England, London, UK

**Acknowledgements** The authors acknowledge technical support from Paul Loveridge, Ana Soriano, and Helen Hughes (PHE Real-time Syndromic Surveillance Team) and support from: NHS 111 and NHS Digital for their assistance and support with the NHS 111 system; OOH providers submitting data to the GPOOH and Advanced; TPP and participating SystmOne practices and University of Oxford, ClinRisk, EMIS and EMIS practices submitting data to the QSurveillance database.

**Contributors** KEM: responsible for conception and design of study, analysis and interpretation of data and drafting the manuscript. FJC-G, RAM, GS and AJE: significant input in the conception and design of study, analysis and interpretation of data and drafting the manuscript and critically reviewed the final version of the work. PRH, JR, GS, SdeL, AY, RP: critically reviewed the final version of the work. IRL: significant input in the conception and design of study, analysis and interpretation of data and drafting the manuscript and critically reviewed the final version of the work. All authors approved the final version of the manuscript.

**Funding** This study was funded by the National Institute for Health Research Health Protection Research Unit (NIHR HPRU) in Emergency Preparedness and Response, a partnership between Public Health England, King's College London and the University of East Anglia. The views expressed are those of the author(s) and not necessarily those of the NIHR, Public Health England or the Department of Health and Social Care.

**Map disclaimer** The depiction of boundaries on this map does not imply the expression of any opinion whatsoever on the part of BMJ (or any member of its group) concerning the legal status of any country, territory, jurisdiction or area or of its authorities. This map is provided without any warranty of any kind, either express or implied.

**Competing interests** None declared.

**Patient consent for publication** Not required.

**Provenance and peer review** Not commissioned; externally peer reviewed.

**Data availability statement** The data are not publicly available due to data sharing agreement with providers.

**ORCID iDs**
Kirsty E Morrison http://orcid.org/0000-0002-1564-2319
Felipe J Colón-González http://orcid.org/0000-0002-9671-3405
Roger A Morbey http://orcid.org/0000-0001-8543-477X
Paul R Hunter http://orcid.org/0000-0002-5608-6144
Simon de Lusignan http://orcid.org/0000-0001-5613-6810
Richard Pebody http://orcid.org/0000-0002-9069-2885
Gillian Smith http://orcid.org/0000-0002-4257-0568
Alex J Elliot http://orcid.org/0000-0002-6414-3065
Iain R Lake http://orcid.org/0000-0003-4407-5357

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
