## [Reviewer comments · BMJ Open]

ARTICLE DETAILS

TITLE (PROVISIONAL)	Demographic and socioeconomic patterns in healthcare-seeking behaviour for respiratory symptoms in England; A comparison with non-respiratory symptoms and between three healthcare services
AUTHORS	Morrison, Kirsty; Colón-González, Felipe; Morbey, Roger; Hunter, Paul; Rutter, Judith; Stuttard, Gareth; de Lusignan, Simon; Yeates, Alex; Pebody, Richard; Smith, Gillian; Elliot, Alex; Lake, Iain

VERSION 1 – REVIEW

REVIEWER	Claudia Flexeder Helmholtz Zentrum München, Germany
REVIEW RETURNED	01-Apr-2020

GENERAL COMMENTS	This study evaluated the demographic and socioeconomic patterns in healthcare-seeking behaviour for respiratory symptoms and diseases in England. Associations between contact rates and the patient's age, sex and deprivation index were analysed. To explore differences in healthcare contact rates, three different healthcare services were evaluated and in addition respiratory contacts and non-respiratory contacts were compared. It could be observed that females had higher contact rates compared to males. Furthermore; males in the younger age groups and those from more deprived areas had increased numbers of contacts. One of the strengths of this study is the large number of about 13 million respiratory contacts. The paper contributes with interesting results to the previous literature. However, some of the statistical analyses have to be described more detailed and some points need to be clarified and further discussed. General comments: • The authors conclude in the abstract that the observed demographic patterns for contact rates are similar to those observed in acute systems. The conclusion should be phrased more carefully as no acute source of data (such as hospital admissions) have been taken into account in this study to support this overall conclusion.• The contacts to the healthcare services are referring to the years 2015 and 2016. Is more detailed information on the time of the year when the healthcare services were contacted available to investigate potential seasonal patterns in healthcare-seeking behaviour? Is there a possibility to take into account whether multiple contacts per patient occurred over time?• The indicators for respiratory diseases used in this study include upper and lower respiratory tract infections, difficulty breathing etc. However, asthma is included as an indicator but not COPD. Could you please comment more detailed on the selection of respiratory disease indicators?• Appendix 5 shows the results for the final models, conducted
--

separately for the three healthcare services as well as stratified by respiratory vs. non-respiratory contacts, in terms of non-transformed main effects and overall significances for interaction terms. In contrast, Figures 1-3 show standardised main effects and standardised effects sizes for the interaction terms. The corresponding text in the results section refers to both Appendix 5 and Figures 1-3. It is not exactly clear how the standardised effect estimates have been calculated as also effects for the reference categories are shown in the figures. If the standardisation of model predictions, supposed to mean standardisation of the predicted values, have been performed as described in lines 244-246, it is not clear which values have been used for the other potential confounding factors included in the prediction equation alongside with the factor of interest. It might be more plausible for the reader and easier to follow and interpret, if Appendix 5 would also show the effect estimates for the interaction terms and if the results section is more clearly structured.

Specific comments:

Abstract:

- It is mentioned in the abstract (lines 87-88) that aggregated statistics on contacts were computed based on 13 million respiratory contacts. Does this mean that the respiratory contacts were aggregated based on specific characteristics or how has the aggregation process been conducted?

- For the NHS111 service a rate ratio of 37.32 is reported for respiratory contacts in the age group under 1 year. What might be the reason for this quite high rate ratio compared to those for the same age group based on the other services (e.g. RR=18.66 for GPOOH service)?

Introduction:

- It is mentioned in line 144 that higher rates of respiratory diseases are also related to higher smoking rates in more deprived areas. Is there a chance to include smoking behaviour as further potential factor or is this covered by the deprivation index?

Methods:

- It is stated that non-respiratory counts were not available for the GPIH health service (line 191). What is the reason that only respiratory counts are available for this specific healthcare service?

- Table 1 and Table 2 might be interchanged. It is mentioned in line 189 that syndromic indicators are presented in Table 1 and it is described in lines 215-216 which kind of data has been excluded (referring to Table 2). However, the reasons for exclusion are summarised in Table 1 and the indicators for respiratory diseases are presented in Table 2.

- It is stated in lines 199-200 that the postcode district grouped by age and sex was used as a denominator in the analyses of NHS111 and GPOOH contacts. It is not exactly clear how this has been taken into account in the generalised linear mixed models, e.g. whether this was modelled as random effects.

- It is mentioned in lines 243-244 that model overfitting was assessed using k-fold cross-validation and results are shown in Appendix 4. However, more information on the cross-validation approach is needed as it is not clear how the partition in test and training samples have been performed and how many runs have been conducted.

- It could be mentioned in the statistical analyses section (lines 247-248) which exact R version has been used to conduct the analyses.

Results:

- It is mentioned in the results section (lines 271-273) that 8.10% of

	the contacts to NHS111 were respiratory contacts whereas 22.53% of the contacts to GPOOH were respiratory contacts. Might this different distribution of respiratory contacts in the healthcare services have an impact on comparability and interpretation of the results? Maybe some more information could be given and discussed. Tables:  • Table 2 presents the indicators for respiratory diseases for each syndromic surveillance system. For instance, asthma is not covered by NHS111. This might have an effect on the comparability of patterns observed for the different healthcare services as the results are based on different respiratory disease indicators. • There are large differences in the marginal R-squared for NHS111 and GPOOH, but the conditional R-squared is comparable (Appendix 5). Could this affect comparability and interpretation of the results of the different healthcare services? Maybe some more information could be added. • Deprivation quintiles are included in the regression models for NHS111 and GPOOH but not for GPIH. The same for % urban. Is there a specific reason for excluding these factors in the GPIH model? Figures:  • Some of the plots (Figures 1-3) are difficult to read as some of the confidence intervals are quite narrow.
--	--

REVIEWER	Dorte Jarbol Department of Public Health, University of Southern Denmark, Denmark
REVIEW RETURNED	08-Apr-2020

GENERAL COMMENTS	Demographic and socioeconomic patterns in healthcare seeking behaviour for respiratory symptoms in England; A comparison with non-respiratory symptoms and between three healthcare services The study explores associations between contact rates with respiratory symptoms to three different healthcare services and the patient's age, gender, and deprivation. Results will be compared between healthcare services, and with non-respiratory contacts to fully explore how these contacts differ by service and illness The study is presented as an ecological study The setting is: Respiratory contacts to three healthcare services in England (NHS111; a telehealth helpline, and GPIH; family doctor services in-hours and GPOOH; unscheduled care) in 2015 and 2016. Overall the study design and overall purpose is difficult to understand as a reviewer outside England. The background introduce to use of syndromic data, which is not further explained. The rationale for the study is unfortunately not motivated in a way that it is possible to read how the results should be interpreted and used. No hypotheses are presented Syndromic indicators; It is not clear what is meant and how they are defined. It is explained that syndromic indicators, classified as respiratory are presented in Table 1. Indicators for respiratory
---

	diseases are however presented in Table 2, as far as I can see. Are they based on diagnoses? ICPC / ICD10 or other classifications? It seems that the authors have mixed up Table 1 and 2. The results are sufficiently presented and discussed, though still I am not sure about how these results adds to the existing knowledge in the research field and their usefulness outside the nation. The healthcare seeking behavior outside England is not referred.
--	---

VERSION 1 – AUTHOR RESPONSE

Editor and Reviewer 1 comments:

General comments:

The authors conclude in the abstract that the observed demographic patterns for contact rates are similar to those observed in acute systems. The conclusion should be phrased more carefully as no acute source of data (such as hospital admissions) have been taken into account in this study to support this overall conclusion.

Author response: Thank you for pointing this out. We have clarified the abstract to make it clear that our findings are similar to those observed in acute healthcare services described in the literature. (Line: 100; 100)

The contacts to the healthcare services are referring to the years 2015 and 2016. Is more detailed information on the time of the year when the healthcare services were contacted available to investigate potential seasonal patterns in healthcare-seeking behaviour? Is there a possibility to take into account whether multiple contacts per patient occurred over time?

Author response: Our broad aim of this research was to investigate the demographic patterns in healthcare seeking behaviours for respiratory contacts to the three healthcare services; we feel that investigating the seasonal patterns would have increased hugely the complexity of the paper, but would be an interesting topic for future research.

Author response: Unfortunately, we were unable to account for multiple contacts per patient due to the aggregate nature of the surveillance systems the data were obtained from. However, we have included a sentence in the limitations to acknowledge this point. (Lines: 559-561; 483-485)

The indicators for respiratory diseases used in this study include upper and lower respiratory tract infections, difficulty breathing etc. However, asthma is included as an indicator but not COPD. Could you please comment more detailed on the selection of respiratory disease indicators?

Author response: The indicators were chosen to be as comparable as possible. However, as we are investigating three different systems with different coding practised, there will of course be some differences. This is acknowledged in the paper lines: 261-264 & 551-555; 227-230 & 476-479. To aid clarity we have also include in the paper further definitions of the healthcare systems used, and the indicators used as well as the source of their coding (Table 1). We have also included further descriptions of the indicators used in Appendix 1.

Author response: In relation to the specific question on COPD, the GP datasets do not collect indicator data on COPD. The “asthma” indicator reflects symptoms such as an asthma attack but is not a clinical diagnosis of asthma.

Appendix 5 shows the results for the final models, conducted separately for the three healthcare services as well as stratified by respiratory vs. non-respiratory contacts, in terms of non-transformed main effects and overall significances for interaction terms. In contrast, Figures 1-3 show standardised main effects and standardised effects sizes for the interaction terms. The corresponding

text in the results section refers to both Appendix 5 and Figures 1-3. It is not exactly clear how the standardised effect estimates have been calculated as also effects for the reference categories are shown in the figures. If the standardisation of model predictions, supposed to mean standardisation of the predicted values, have been performed as described in lines 244-246, it is not clear which values have been used for the other potential confounding factors included in the prediction equation alongside with the factor of interest. It might be more plausible for the reader and easier to follow and interpret, if Appendix 5 would also show the effect estimates for the interaction terms and if the results section is more clearly structured.

Author response: The referee is correct, and the standardised values are calculated using the predicted values, hence we have results for the reference categories. We agree that the description of the standardisation method could be more clearly described and we have included further details of the standardisation process (Lines: 334-348; 299-303).

Author response: We decided not to include the interaction terms in Appendix 5 because we feel they can be visualised better through plotting.

Author response: The referee also asks for our results to be structured in a clearer way and we have been through these again to improve clarity. (Lines: 376-441; 330-394)

Abstract:

It is mentioned in the abstract (lines 87-88) that aggregated statistics on contacts were computed based on 13 million respiratory contacts. Does this mean that the respiratory contacts were aggregated based on specific characteristics or how has the aggregation process been conducted?

Author response: The respiratory contacts were aggregated by summing the syndromic indicators for each healthcare service (grouped by location, age, and gender). The 13 million contacts refer to the three healthcare systems combined. We feel that the best location to detail of this aggregation process is in the methods and we have provided this (Lines:263-267; 231-234)

For the NHS111 service a rate ratio of 37.32 is reported for respiratory contacts in the age group under 1 year. What might be the reason for this quite high rate ratio compared to those for the same age group based on the other services (e.g. RR=18.66 for GPOOH service)?

Author response: We have addressed this point in the discussion (lines: 490-496; 417-422). The differences observed in contacts between these two services could be due to the type of service they offer. NHS111 is a non-emergency help line which can give advice on whether to, and where to seek further medical help; GPOOH is an out of hours service for family doctors and is for urgent situations outside of normal working hours that require a patient to consult with a GP. It could be that parents with young children (<1 year) might require advice from the services of NHS111 where they are not certain where to go or what to do about a situation, whereas they will only go to GPOOH services if there is a more urgent problem. It may also reflect the acute nature of respiratory illnesses in very young children.

Introduction:

It is mentioned in line 144 that higher rates of respiratory diseases are also related to higher smoking rates in more deprived areas. Is there a chance to include smoking behaviour as further potential factor or is this covered by the deprivation index?

Author response: The deprivation index does not include smoking, but deprivation and smoking are highly correlated (1). To avoid issues with collinearity, it was decided to include this broad index rather than more specific variables, such as smoking, and other factors associated with both respiratory illness and deprivation such as housing quality or overcrowding. In the methods section we included a further description of the deprivation index and the indices used to calculate. We have also clarified why this broad index was chosen rather than more specific factors. (Lines: 280-284; 246-250)

It is stated that non-respiratory counts were not available for the GPIH health service (line 191). What is the reason that only respiratory counts are available for this specific healthcare service?

Author response: Unfortunately, the total number of consultations is not available from the GPIH system. The number of contacts for specific indicators (such as those used in this research) are provided as these are essential for the public health aims of the surveillance system.

Table 1 and Table 2 might be interchanged. It is mentioned in line 189 that syndromic indicators are presented in Table 1 and it is described in lines 215-216 which kind of data has been excluded (referring to Table 2). However, the reasons for exclusion are summarised in Table 1 and the indicators for respiratory diseases are presented in Table 2.

Author response: We apologise for this mistake; the tables were labelled incorrectly and have now been corrected.

It is stated in lines 199-200 that the postcode district grouped by age and sex was used as a denominator in the analyses of NHS111 and GPOOH contacts. It is not exactly clear how this has been taken into account in the generalised linear mixed models, e.g. whether this was modelled as random effects.

Author response: Postcode district was included in the model as a random effect. This has been clarified in the methods section (Lines: 317-318; 283-284). We have also included the equations for the two models used in the analysis in Appendix 5 to clarify the statistical methods used.

It is mentioned in lines 243-244 that model overfitting was assessed using k-fold cross-validation and results are shown in Appendix 4. However, more information on the cross-validation approach is needed as it is not clear how the partition in test and training samples have been performed and how many runs have been conducted.

Author response: We agree that further clarification is needed. We have included a further description of the k-fold cross-validation methods in the appendix. (Lines: 339-342; 295-298)

It could be mentioned in the statistical analyses section (lines 247-248) which exact R version has been used to conduct the analyses.

Author response: We agree. The statistical analysis was conducted in R version 3.5.2. This has been included in the methods section. (Line: 349; 304)

Results:

It is mentioned in the results section (lines 271-273) that 8.10% of the contacts to NHS111 were respiratory contacts whereas 22.53% of the contacts to GPOOH were respiratory contacts. Might this different distribution of respiratory contacts in the healthcare services have an impact on comparability and interpretation of the results? Maybe some more information could be given and discussed.

Author response: The different proportions of respiratory contacts between the two systems reflect the different function of the services, and the nature at which each service would be contacted (with GPOOH receiving more acute medical issues). Further detail of each service is provided in Table 1. We agree it is important to discuss why there is this difference in proportions and have included a sentence in the results to refer to this point (Lines: 384-386; 339-343)

Table 2 presents the indicators for respiratory diseases for each syndromic surveillance system. For instance, asthma is not covered by NHS111. This might have an effect on the comparability of patterns observed for the different healthcare services as the results are based on different respiratory disease indicators.

Author response: The three healthcare services have different coding practices and systems therefore they will not be directly comparable. But these indicators were chosen based on their ability to detect acutely presenting respiratory illnesses and therefore there will be a high degree of comparability. We

have tried to address this issue by providing further detail on how they were chosen (Lines:260-263; 227-230), further definitions of the services (Table 1) and indicators (Appendix 1) and discussing it in the limitations section (Lines: 551-554; 476-479)

There are large differences in the marginal R-squared for NHS111 and GPOOH, but the conditional R-squared is comparable (Appendix 5). Could this affect comparability and interpretation of the results of the different healthcare services? Maybe some more information could be added.

Author response: We have included further information of this in our results (lines: 393-403; 347-357). The difference in marginal R-Squared is because the explanatory variables (deprivation, age, gender and % urban) explain less of the variation in the GPOOH data compared to the NHS111 data. The conditional R-Squared is comparable because the remaining variation can be explained by unobserved factors at the random effect level (Postcode district and upper tier local authority). The difference between the marginal and conditional R-squared in the GPOOH models is likely due to the uncertainty of the underlying study population (we tried to remove areas of poor coverage, but some areas may have remained in the analysis), with this uncertainty explained at the geographical levels (random effects). The NHS111 system has a much higher level of coverage (between 90-100%), therefore did not have as much uncertainty in the underlying population. We argue this will not impact the final results, but the uncertainty in population will increase the confidence intervals around our estimates.

Deprivation quintiles are included in the regression models for NHS111 and GPOOH but not for GPIH. The same for % urban. Is there a specific reason for excluding these factors in the GPIH model?

Author response: Data for GPIH was only available at regional level, which is a very large geographical level with only nine areas. When deprivation and % urban data was aggregated to this level there was very little variation. Hence it was best to not include these variables in the analysis, as non-significant estimates would likely be due to the aggregation of the data to this high level of geography, and not due to a lack of relationship with respiratory contacts. We have clarified. (lines: 311-315; 277-282):

Figures:

Some of the plots (Figures 1-3) are difficult to read as some of the confidence intervals are quite narrow.

Author response: We concur on the importance of getting the plots as clear as possible. The narrow confidence intervals observed for the GPIH data are because the underlying study population was known, where it was estimated for NHS111 and GPOOH data. This has resulted in comparably larger confidence intervals for the NHS111 and GPOOH due to the uncertainty. We have redesigned the figures, and feel they are now much clearer (Figures 1,2 and 3).

Reviewer 2 comments:

Overall, the study design and overall purpose is difficult to understand as a reviewer outside England.

Author response: It is important for the wider relevance of this study to be clearer and we have tried to address this point and enhance the wider applicability of the research. Therefore, we have included further definitions of the healthcare services used in this analysis and the indicators used to provide clarity for those not based in England (Table 1). The study design and purpose has also been clarified in the introduction (Lines: 188-196; 158-166). In the discussion we have also commented upon the wider applicability of the research (lines: 537-542; 461-466).

The background introduces to use of syndromic data, which is not further explained.

Author response: We have further explained our rationale behind the use of syndromic data, and why we are using it in the context of this research (Lines: 188-196; span style="font-family:'Times New Roman'; color:#ff00ff">158-166).

The rationale for the study is unfortunately not motivated in a way that it is possible to read how the results should be interpreted and used. No hypotheses are presented

Author response: We thank the reviewer for highlighting this issue, and in our revised introduction we have stated more clearly the rationale for the study (Lines: 188-196; 158-166). We have not presented a hypothesis for this research (in common with similar papers (2)) but have listed our aims of the research in the introduction.

Syndromic indicators; It is not clear what is meant and how they are defined. It is explained that syndromic indicators, classified as respiratory are presented in Table 1. Indicators for respiratory diseases are however presented in Table 2, as far as I can see. Are they based on diagnoses? ICPC / ICD10 or other classifications? It seems that the authors have mixed up Table 1 and 2.

Author response: We apologise for the mix up in table numbers, this has been corrected.

Author response: We have further defined the services and described the source coding for the indicators (Table 1), and definitions of the indicators used are in Appendix 1.

The results are sufficiently presented and discussed, though still I am not sure about how these results adds to the existing knowledge in the research field and their usefulness outside the nation. The healthcare seeking behaviour outside England is not referred.

Author response: We have been through our discussion and included further references from studies outside of England in the discussion. We have discussed why it is important for countries to undertake similar research as this study (lines: 537-542; 461-466)

1. Hiscock R, Bauld L, Amos A, Platt S. Smoking and socioeconomic status in England: the rise of the never smoker and the disadvantaged smoker. J Public Health (Oxf). 2012 Aug 1;34(3):390–6.

2. Adams NL, Rose TC, Elliot AJ, Smith G, Morbey R, Loveridge P, et al. Social patterning of telephone health-advice for diarrhoea and vomiting: analysis of 24 million telehealth calls in England. Journal of Infection. 2019;78(2):95-100; Available from: <http://www.sciencedirect.com/science/article/pii/S0163445318302810>

VERSION 2 – REVIEW

REVIEWER	Claudia Flexeder Helmholtz Zentrum München, Germany
REVIEW RETURNED	21-Jul-2020
GENERAL COMMENTS	I thank the authors for the thorough revision of the manuscript and addressing all of my comments. I have no further comments or suggestions.
REVIEWER	Dorte Jarbol Research Unit of General Practice, Department of Public Health, University of Southern Denmark
REVIEW RETURNED	11-Jul-2020
GENERAL COMMENTS	The authors have responded to my comments in a sufficient way.